# Synthetic Pyrethroids Exposure and Embryological Outcomes: A Cohort Study in Women from Fertility Clinic

**DOI:** 10.3390/ijerph19095117

**Published:** 2022-04-22

**Authors:** Paweł Radwan, Bartosz Wielgomas, Michał Radwan, Rafał Krasiński, Anna Kilanowicz-Sapota, Renata Banaszczyk, Joanna Jurewicz

**Affiliations:** 1Gameta Health Centre, 7 Cybernetyki St., 02-677 Warsaw, Poland; pradwan@gameta.pl; 2Gameta, Kielce-Regional Science-Technology Centre, 45 Podzamcze St., Chęciny, 26-060 Kielce, Poland; 3Department of Toxicology, Medical University of Gdańsk, 107 Hallera St., 80-416 Gdansk, Poland; bartosz.wielgomas@gumed.edu.pl; 4Department of Gynecology and Reproduction, Gameta Hospital, 34/36 Rudzka St., 95-030 Rzgow, Poland; mradwan@gameta.pl (M.R.); rkrasinski@gameta.pl (R.K.); rbanaszczyk@gameta.pl (R.B.); 5Faculty of Health Sciences, Mazovian State University in Plock, 2 Dabrowskiego Sq., 09-402 Plock, Poland; 6Department of Toxicology, Medical University of Lodz, Muszyńskiego 1, 90-151 Lodz, Poland; anna.kilanowicz@umed.lodz.pl; 7Department of Chemical Safety, Nofer Institute of Occupational Medicine, 8 Teresy St., 91-348 Lodz, Poland

**Keywords:** embryological outcomes, urinary pyrethroids concentrations, environmental exposure, IVF treatment

## Abstract

Pyrethroids exposure has been associated with adverse reproductive outcome. However, there is no study that explores the effect of environmental exposure and embryological outcomes. This question was addressed in a prospective cohort of couples undergoing fertility treatment. The study aims to assess the association between urinary metabolites of synthetic pyrethroids and embryological outcomes (MII oocyte count, top quality embryo, fertilization and implantation rate). We included 450 women aged 25–45 undergoing assisted reproductive technology (ART) cycle at Infertility Clinic in Poland. Urine samples were collected at the time of fertility procedure(s) to assess four urinary synthetic pyrethroids concentrations (3-phenoxybenzoic acid (3PBA), cis-3-(2,2-dichlorovinyl)-2,2-dimethylcyclopropane carboxylic acid (cis-DCCA), trans-3-(2,2-dichlorovinyl)-2,2-dimethylcyclopropane carboxylic acid (trans-DCCA), cis-2,2-dibromovinyl-2,2-dimethylocyclopropane-1-carboxylic acid (DBCA)) using validated gas chromatography ion-tap mass spectrometry and calculated for each cycle-specific metabolite. To evaluate the effect of environmental exposure to synthetic pyrethroids and embryological outcomes (methaphase II (MII) oocyte yield, top quality embryo, fertilization rate, implantation rate), multivariable generalized linear mixed analyses with random intercepts were prepared. Urinary 3-PBA concentrations decrease MII oocyte count (*p* = 0.007) in the fourth quartile (>75 percentile) compared to women in the first quartile (≤25 percentile). Additionally, when 3-PBA was treated as continuous variable, the negative association between exposure to pyrethroids and MII oocyte count was also observed (*p* = 0.012). Exposure to other pyrethroid metabolities (CDCCA, TDCCA, DBCA) was not related to any of the examined embryological outcomes. Exposure to synthetic pyrethroids may be associated with poorer embryological outcome among couples seeking fertility treatments. As this is the first study on this topic, the results need to be confirmed in further studies.

## 1. Introduction

A class of insecticides, synthetic pyrethroids, have widely, continuously been used to control insect pests in numerous commercial formulations in recent years. Synthetic pyrethroids are used worldwide in agricultural, residential, public, and veterinary area for insect control [1]. Their chemical structure is based on naturally occurring pyrethrins, which are found in the flowers of Chrysanthemum cineraraefolum. The basic pyrethroid structure consists of an acid and an alcohol moiety, with an ester bond. Pyrethroids have been suspected as endocrine-disrupting chemicals (ECDs) and listed as ECDs because of their hormone-like effects by the Environmental Protection Agency in the US [2,3]. One implication of the extensive and increasing use of synthetic pyrethroid insecticides is widespread exposure among general population. Humans can be exposed primarily via the consumption of pesticide-residue-contaminated food, but also via the inhalation or ingestion of contaminated dust after application or through dermal absorption, inhalation, or ingestion of veterinary formulations to control pests [4].

As pyrethroids are nonpersistent insecticides, they are quickly metabolized and eliminated as non-specific or specific metabolites [4]. Insecticides undergo metabolic processes in two phases. In phase I reactions, the metabolic enzymes change the insecticide compounds to more water-soluble products than the originals via reduction, oxidation, or hydrolysis, whereas in phase II reactions, the conjugation with hydrophilic molecules occurs to increase the water solubility for excretion in urine [5]. The plasma half-life is less than 8 h through hydrolysis and oxidation [5].

Because of the numerous applications and increasing worldwide use of pyrethroids, more and more attention is being paid to the risk of exposure to humans. Pyrethroids have been linked to the disruption of the endocrine system, which can adversely affect reproduction and sexual development and interfere with the immune system [1].

Studies regarding the effect of environmental exposure to pyrethroids on reproduction are limited. Epidemiological studies found poorer semen quality, sperm DNA damage, and sperm aneuploidy and negative effect on hypothalamic-pituitary-gonadal glands associated with exposure to synthetic pyrethroids [3,6,7].

Exposure to pyrethroids affect female reproductive functions, decrease steroid hormones levels, disorder estrous cycles, and restrain follicle cells. This causes poor oocyte maturation and competency, meiotic maturation of oocytes, embryonic defects, and poor IVF outcomes in animal studies [8,9,10,11,12,13,14]. This indicates that exposure to these substances could affect the fertility of the animal. Human studies that have assessed the effect of synthetic pyrethroids environmental exposure on female fertility are rare and mainly focused on ovarian reserve or time to pregnancy. According to our knowledge, no studies examine the effect of exposure and early IVF outcomes.

In African women, decreases in the level of anti-Mullerian hormone (AMH) were observed [15]. Primary ovarian insufficiency was associated with urinary levels of 3-PBA (3-phenoxybenzioc acid) [16]. A study performed in Poland found that synthetic pyrethroids affect ovarian reserve parameters (AMH, intral follicle count and FSH (follicle stimulating hormone) levels) [17]. Longer TTP and infertility were associated with a urinary concentration of 3-phenoxybenzoic acid in a general population of couples planning pregnancy in China [4].

This is the first study to assess environmental exposure to synthetic pyrethroids and embryological outcome among women undergoing IVF treatment.

The association between preconception environmental exposure to synthetic pyrethroids and early embryological outcomes has never been examined. Thus, this study aims to assess the association between urinary metabolites of synthetic pyrethroids and embryological outcomes (MII oocyte count, top quality embryo, fertilization, and implantation rate). These sensitive early embryological outcomes, all of which are critical to human reproduction, rely on very carefully balanced hormonal triggers and have been shown to be susceptible to the endocrine-disrupting properties of synthetic pyrethroids in animal studies. Identifying modifiable factors, such as environmental exposures, that can predict human fertility has become a major clinical and public health concern.

## 2. Materials and Methods

### 2.1. Study Participants

This is a prospective cohort study of couples undergoing fertility treatments for examining the environmental exposure which may affect fertility [18]. Women between the ages 25–45 years old with maximum body mass index (BMI) threshold of 40 kg/m^2^ were recruited and eligible for inclusion in the study as previously described Radwan et al., 2020 [18]. Chlamydia infection and thyroid disfunction (TSH > 2.5 µU/mL) exclude the potential participants from the study. The Bioethical Committee of Nofer Institute of Occupational Medicine, Lodz, Poland approved study protocols and all participants submitted consent prior to study initiation. We recruited 450 women, between 2017–2019, who underwent at least one fresh in vitro fertilization cycle (*n* = 674 IVF cycles). At enrollment, height and weight were measured by trained research study staff to calculate body mass index (BMI (kg/m^2^)), and data on demographics, medical and reproductive history, and lifestyle characteristics and occupational factors were collected via questionnaire. Women were then followed prospectively through their ART cycles until failure or live birth.

### 2.2. Embryological Outcomes and Hormone Level Measures

All study participants underwent a standard infertility work-up. Infertility diagnosis was assigned according to definitions of the Society for Assisted Reproductive Technology (SART 2015). To stimulate ovulation, long agonist or short antagonist protocol was used as previously described [18].

To identify and isolate cumulus-oocyte complexes (COCs), follicular fluid was evaluated under a microscope as previously described [17]. The classifications of oocytes were as follows: germinal vesicle (GV), methaphase I (MI), methaphase II (MII), or degenerated. After sufficient follicular development, women typically underwent oocyte retrieval on cycle day 14. Embryo transfer occurred 3 to 5 days after oocyte retrieval. Grade A and B notes were used for top quality embryos. The implantation rate was defined as the percentage of embryos with successful implantation compared to the number of embryos transferred.

Blood samples were obtained for measurement of basic hormone levels at the beginning of the IVF cycle. FSH, estradiol (E2), luteinizing hormone (LH), and progesterone were assessed in serum using chemiiluminescence immunoassay. An enzyme-linked immunoabsorbent method using utilizing commercially available Gen-II ELISA kits according to the manufacturer’s instruction (Beckman Coulter Inc., Brea, CA, USA) was used to assess the AMH levels in serum.

### 2.3. Urine Collection and Pyrethroids Metabolities Quantification

Women provided one (32%) or two (68%) spot urine samples per IVF cycle at the beginning and in the middle of IVF cycle in a polypropylene specimen cup. Specific gravity (SG) was measured at room temperature using a handheld refractometer calibrated with deionized water before each measurement. A validated gas chromatography ion-tap mass spectrometry was used to quantify the urinary concentrations of pyrethroid metabolities, including 3-phenoxybenzoic acid (3PBA), cis-3-(2,2-dichlorovinyl)-2,2-dimethylcyclopropane carboxylic acid (cis-DCCA), trans-3-(2,2-dichlorovinyl)-2,2-dimethylcyclopropane carboxylic acid (trans-DCCA), and cis-2,2-dibromovinyl-2,2-dimethylocyclopropane-1-carboxylic acid (DBCA) with LOD 0.1 µg/L for each metabolites [16]. Urinary pyrethroid metabolite concentrations below the limit of detection (LOD) were replaced with LOD divided by the square root of 2. Cycle-specific urinary concentration was assessed as the geometric mean of the pyrethroid metabolite concentrations from two spot urine samples except for cycles with only one urine sample. Urinary synthetic pyrethroid metabolities were also presented as a sum of measured metabolites.

### 2.4. Statistical Analysis

Demographic and reproductive characteristics were summarized based on information from the first study cycle using arithmetic means ± standard deviations (SDs) or counts (%). Pyrethroid metabolities concentrations were summarized using percentiles and arithmetic, geometric means ± standard deviations (SDs).

3-PBA urinary concentrations were categorized into quartiles (>25 to 50th, >50 to 75th, >75th), with the lowest quartiles considered as the reference group (<LOD to 25th). The concentrations of CDCCA, TDCCA, and DBCA were treated as <LOD (reference group) and >LOD due to low detection frequency 34.3%, 45% and 22.1%, respectively. Pyrethroid metabolite urinary concentration was also treated as continuous variable. Spearman statistics were used for correlation between urinary pyrethroid metabolites. Multivariable generalized linear mixed analyses with random intercepts were used to evaluate the association of urinary perythroid metabolite levels and studied embryological outcomes, taking into account potential confounding factors. The variables considered potential confounders included factors previously related to early reproductive endpoints, and factors associated with urinary pyrethroids metabolites and reproductive outcomes in this study. Final models were adjusted for urinary dilution (SG), age, body mass index (BMI), smoking status, infertility diagnosis, protocol type, and antral follicle count (AFC). Statistical analyses were conducted with R statistical software (Version 3) [19]. Statistical tests were two-tailed, and all *p*-values < 0.05 were regarded as statistically significant.

## 3. Results

The analysis sample consisted of 450 women with a mean age of 34.0 years and BMI of 23.2 kg/m^2^ who mostly had higher education (65.11%); 8% were current smokers, and 54.44% drank none or less than one drink per week (Table 1). The most common infertility diagnosis was male factor (38%). Most female participants underwent a luteal phase GnRH agonist stimulation protocol (56%) during their IVF cycles. The mean level of AFC (in both ovaries) was 12.54 ± 7.21. The level of examined hormones, FSH, E2, LH, progesterone, and peak estradiol level, was 6.21 ± 1.13 IU/l, 92.78 ± 15.78 pg/mL, 5.34 ± 3.24 IU/l, 0.99 ± 1.58 ng/mL, and 2608.74 ± 1614.63 ng/mL, respectively (Table 1).

The embryological outcome measures are presented in Table 1. The mean ± SD number of oocytes retrieved, MII oocytes, embryos available, top-quality embryos, and embryos transferred was 9.98 ± 7.22, 13.21 ± 7.44, 6.28 ± 1.54, 2.01 ± 2.48, and 1.95 ± 0.16, respectively. The rates of fertilization, implantation and clinical pregnancy were 80% ± 21, 38.16% ± 2.79 and 43.12 ± 7.89 (Table 1).

A total of 739 urine samples were collected from the 450 women undergoing 674 IVF cycles and are included in this analysis (Table 2). The detection frequencies for urinary concentrations of pyrethroids metabolites ranged from 22.11 to 68.13%. The most frequently detected pyrethroid metabolite was 3-PBA with geometric mean concentration ± SD 0.35 ± 2.66 μg/L (0.22 ± 2.68 μg/L SG-adjusted) (Table 2). CDCCA and TDCCA were detected in about 34% and 45% with mean urinary level 0.29 ± 2.18 μg/L (0.11 ± 2.24 μg/L SG-adjusted) and 0.43 ± 2.48 μg/L (0.32 ± 2.76 μg/L SG-adjusted) (Table 2). DBCA was quantified in 22% of samples with mean concentration 0.28 ± 2.51 μg/L (0.18 ± 2.55 μg/L SG-adjusted). The GM ± SD sum of measured metabolites was 0.77 ± 2.46 μg/L (0.75 ± 2.28 μg/L SG-adjusted) (Table 2).

Urinary concentrations of all the examined pyrethroid metabolites were significantly correlated with each other (*p* < 0.001) (Table 3).

In models adjusted for SG, BMI, age, smoking, infertility diagnosis, and AFC, protocol type urinary 3-PBA concentrations were negatively associated with MII oocyte count (*p* = 0.007) in the fourth quartile (>75 percentile) compared to women in the first quartile (≤25 percentile) (Table 4). Additionally, when 3-PBA was treated as a continuous variable, a decrease in MII oocyte count was also observed (*p* = 0.012). Exposure to 3-PBA was not related to top quality embryo, fertilization rate and implantation rate of top-quality embryo, fertilization rate, or implantation. Additionally, exposure to other pyrethroid metabolities (CDCCA, TDCCA, DBCA) was not associated with any of examined embryological outcomes (MII oocyte count, top-quality embryo, fertilization rate, or implantation) (Table 4).

## 4. Discussion

In the current analysis, we evaluated the association between urinary synthetic pyrethroid metabolities and embryological outcomes of women undergoing IVF treatment. Only urinary 3-PBA levels decreased MII oocyte count. Other examined parameters (top-quality embryos, fertilization, and implantation rate) were not related to 3-PBA levels. Additionally, the remaining assessed metabolites (TDCCA, CDCCA, and DBCA) were not associated with the examined embryological outcomes. As this is a preliminary study, which evaluates environmental exposure to synthetic pyrethroids and embryological outcome, it is difficult to compare the results.

Most of the epidemiological studies on the effect of exposure to synthetic pyrethroids and reproductive health are focused on semen quality parameters including DNA damage and aneuploidy and the level of hormones associated with the hypothalamic-pituitary-gonadal axis [3,6,7].

Additionally, previously published epidemiological studies suggest that pesticide exposures have adverse effects on fertility assessed by time to pregnancy (TTP). In Denmark, women working in flowery greenhouses who work on cultivation without gloves had lower fertility (FOR = 0.697 (95% CI: 0.46, 0.98)) than women who always used gloves [20]. In Colombia, women who had been employed in the flower production industry for ≥2 y had a significantly longer TTP (FOR = 0.73 (95% CI: 0.63, 0.84)) than other workers (Idrovo et al. (2005)) [21]. On the other hand, studies conducted in the Netherlands [22] and Italy [23] found no associations between pesticide exposure and TTP, but exposure to pesticides in these studies was self-reported, not based on real measurements of pesticide concentration in urine. Presumed exposure to organophosphate and pyrethroid metabolities were assessed in urine in the study performed in China [4] where the urinary concentration of 3-phenoxybenzoic acid was associated with longer TTP and infertility.

A decreased level of AMH associated with pyrethroids exposure was observed in African women [14]. Urinary levels of 3-PBA were associated with primary ovarian insufficiency [15]. An impact on ovarian reserve parameters (AMH, antral follicle count, FSH) was also observed in the case of environmental exposure to synthetic pyrethroids [16].

Animal studies suggest that synthetic pyrethroids exposure can inhibit steroid hormones, restrain the growth of follicles, and damage ovarian corpus luteum cells, which might further contribute to decreased fertility [8,9,24,25]. Cypermetrin, deltametrin, and fenvalerate were found to significantly affect meiotic maturation. The effects depended on the stage of competence of oocytes [14]. In fully grown pig oocytes with full meiotic competence, maturation in vitro was delayed. This demonstrates a significant effect of pyrethroids maturation of mammalian oocytes under in vitro conditions. This indicates that exposure to these substances could affect the fertility.

On the other hand, no studies assess embryological outcome among women undergoing IVF treatment.

A potential biological mechanism that might explain association of pyrethroids exposure with oocyte quality and embryo development may be derived from animal studies. Exposure to those compounds may cause oxidative stress [11] and impact oocyte quality, oocyte dysmorphisms, and poor competence for development into a good embryo for implantation. Dysregulating embryonic genome activation and embryonic metabolism and negative impacts on embryonic development, which may consequently give poor clinical outcomes, were also observed after exposure to pesticides [26].

The median and 95th percentile were comparable in our study in case of 3-PBA (0.30 ng/mL, 2.10 ng/mL, respectively) and in the National Health and Nutrition Examination Survey 2017 (0,25 ng/mL, 2.30 ng/mL, respectively) [27]. CDCCA, TDCCA, and DBCA urinary levels were higher in the current study (75th percentile was: 0.33 ng/mL, 0.66 ng/mL, 0.48 ng/mL, respectively) than in the US (75th percentile was: 0.18 ng/mL, 0.56 ng/mL, <LOD, respectively).

The current analysis is the first human study to assess exposure to synthetic pyrethroids and embryological outcomes. The strengths of our study include performing the study in the same center, using the same standardized protocol. Detailed questionnaire data on demographics, medical, and lifestyle risk factors allowed for control in the statistical analysis. Additionally, all study participants provided at least one urine sample per cycle, which, in the case of nonpersistent chemicals with short half-lives, is very important to confirm the exposure. The next advantage is the big sample size, with 450 women who provided 730 urine samples in 674 cycles. Our study has also several limitations. It may be difficult to generalize the results to the general population, as the study was conducted among women seeking infertility treatment. Additionally, the study is not able to show the mechanism of the observed associations, but the epidemiological study, by nature, is not designed to study the mechanisms.

## 5. Conclusions

In conclusion, 3-PBA may affect MII oocyte count, which suggests that exposure to synthetic pyrethroids may affect embryological outcomes. These findings highlight the importance of further examination of the current findings. As the environmental exposure to synthetic pyrethroids is related to poorer embryological outcomes, mitigating pyrethroids exposure should be considered. Unnecessary exposure to pyrethroids should be avoided: they should only be used when there is a need, and levels of exposure and amounts used should be kept to a minimum. As diet plays an important role in the presence of pyrethroid metabolites, the identification of these products helps to avoid pyrethroid exposure.

## Figures and Tables

**Table 1 ijerph-19-05117-t001:** Baseline characteristics and IVF outcome among the study population *n* = 450.

Variables	
PER WOMEN	
**Education** (*n* (%))	
Vocational	13 (2.89)
Secondary	144 (32.00)
Higher	293 (65.11)
**Age (years)** (*n* (%))	
24–30	81 (18.00)
31–39	342 (76.00)
40–44	27 (6.00)
Mean ± SD	31.28 ± 3.52
**BMI (kg/m^2^)** (*n* (%))	
<18.5	26 (5.78)
18.5–24.9	261 (58.00)
25–29.9	135 (30.00)
30–40	28 (6.22)
Mean ± SD	23.19 ± 2.67
**Current smoking** (*n* (%))	
No	414 (92.00)
Yes	36 (8.00)
**Alcohol use**	
None or <1 drink/week	245 (54.44)
1–3 drinks /week	198 (44.0)
Everyday	7 (1.56)
**Duration of couple’s infertility (years)** (*n* (%))	
1–2	34 (7.56)
2–3	121 (26.89)
3–5	131 (29.11)
>5	164 (36.44)
**PER CYCLE**	
**Initial infertility diagnosis** (*n* (%))	
Male factor	171 (38.0)
Idiopathic	139 (30.89)
Endometriosis	62 (13.78)
Ovarian factor	21 (4.67)
Tubal factor	46 (10.22)
Missing data	11 (2.44)
**Stimulation protocol**	
Long GnRH agonist protocol	297 (44.10%)
GnRH-antagonist protocol	377 (55.90%)
**AFC** (*n*) (mean ± SD)	12.54 ± 7.21
**FSH (IU/l)** (mean ± SD)	6.21 ± 1.13
**E2 peak (ng/mL)** (mean ± SD)	2608.74 ± 1614.63
**E2 (pg/mL)** (mean ± SD)	92.78 ± 15.78
**Progesterone** (ng/mL) (mean ± SD)	0.99 ± 1.58
**LH (IU/l)** (mean ± SD)	5.34 ± 3.24
**AMH** (ng/mL) (mean ± SD)	11.19 ± 1.22
**Number of oocytes retrieved (COCs)** (mean ± SD)	13.21 ± 7.44
**Number of MII oocytes** (mean ± SD)	9.98 ± 7.22
**Fertilization rate (%)** (mean ± SD)	80 ± 21
**No. of cleavage embryos available** (mean ± SD)	6.28 ± 1.54
**No. of top quality embryos** (mean ± SD)	2.01 ± 2.48
**No. of embryos transferred** (mean ± SD)	1.95 ± 0.16
**Clinical pregnancy rate (%) per cycle** (mean ± SD)	43.12 ± 7.89
**Implantation rate (%) per embryo** (mean ± SD)	38.16 ± 2.79

SD—standard deviation; AMH—Anti-Müllerian hormone; AFC—antral follicle count; FSH—follicle-stimulating hormone; E2—estradiol, LH—Luteinizing hormone.

**Table 2 ijerph-19-05117-t002:** Distribution of urinary concentration of metabolites od synthetic pyrethroids among 450 women contributing 739 urine samples.

Metabolites of Synthetic Pyrethroids in Urine	Statistics							
	A Mean (SD)	G Mean (SD)	LOD	Q25	Median	Q75	Q95	Detection Frequency (%)
ng/mL								
CDCCA	0.47 (1.39)	0.29 (2.18)	0.1	0.15	0.21	0.33	1.34	34.28
TDCCA	1.11 (3.48)	0.43 (2.48)	0.1	0.24	0.36	0.66	2.88	44.95
DBCA	0.56 (1.32)	0.28 (2.51)	0.1	0.14	0.24	0.48	2.21	22.11
3-PBA	0.78 (2.37)	0.35 (2.66)	0.1	0.17	0.30	0.58	2.10	68.13
Sum of measured metabolites	1.09 (2.17)	0.77 (2.46)	0.1	0.32	0.75	1.11	5.12	-
**SG adjusted (ng/mL)**								
CDCCA	0.32 (1.29)	0.11 (2.24)	0.1	0.06	0.11	0.22	1.16	34.28
TDCCA	1.09 (2.18)	0.32 (2.76)	0.1	0.17	0.31	0.69	3.28	44.95
DBCA	0.36 (1.08)	0.18 (2.55)	0.1	0.09	0.17	0.28	2.31	22.11
3-PBA	0.68 (2.15)	0.22 (2.68)	0.1	0.08	0.20	0.46	2.02	68.13
Sum of measured metabolites	0.98 (2.35)	0.75 (2.28)	0.1	0.28	0.74	1.06	4.88	-

A Mean—artimetic mean; G Mean—geometric mean; Q25—25th quartile; Q75—75th quartile; Q95—95th quartile; LOD—limit of detection; 3PBA—3-phenoxybenzoic acid; CDCCA, TDCCA—cis- and trans-3-(2,2-Dichlorovinyl)-1-methylcyclopropane-1,2-dicarboxylic acid; DBCA—cis-2,2-dibromovinyl-2,2-dimethylcyclopropane-carboxylic acid; Per IVF cycle (*n* total = 674), 1 (*n* = 101; 15%) or 2 (*n* = 573; 85%).

**Table 3 ijerph-19-05117-t003:** Spearman correlation between metabolites of synthetic pytethroids.

	CDCCA	TDCCA	DBCA	3-PBA
CDCCA	1	<0.001	<0.001	<0.001
TDCCA	0.71	1	<0.001	<0.001
DBCA	0.31	0.22	1	<0.001
3-PBA	0.68	0.55	0.42	1

Note: above diagonal diagonal-*p* values, Below—correlations; 3PBA—3-phenoxybenzoic acid; CDCCA, TDCCA—cis- and trans-3-(2,2-Dichlorovinyl)-1-methylcyclopropane-1,2-dicarboxylic acid; DBCA—cis-2,2-dibromovinyl-2,2-dimethylcyclopropane-carboxylic acid.

**Table 4 ijerph-19-05117-t004:** The association between urinary metabolites of synthetic pyrethroids concentrations and embriological outcomes among 450 women undergoing 674 cycles.

		MII Oocyte Count	Top Quality Embryo	Fertilization Rate	Implantation
	Categories	coef (95% CI); *p*	coef (95% CI); *p*	coef (95% CI); *p*	coef (95% CI); *p*
CDCCA	Cont.	−0.021 (−0.081; 0.039); 0.234	−0.015 (−0.066; 0.036); 0.287	0.012 (−0.034; 0.059); 0.371	−0.052 (−0.132; 0.028); 0.204
	>LOD	−0.025 (−0.091; 0.041); 0.462	−0.009 (−0.071; 0.052); 0.766	0.034 (−0.202; 0.271); 0.776	−0.072 (−0.307; 0.163); 0.548
TDCCA	Cont.	−0.034 (−0.09; 0.021); 0.549	0.003 (−0.034; 0.04); 0.854	−0.01 (−0.034; 0.014); 0.395	−0.055 (−0.118; 0.008); 0.085
	>LOD	−0.047 (−0.118; 0.024); 0.196	0.003 (−0.048; 0.055); 0.423	0.008 (−0.131; 0.148); 0.91	−0.063 (−0.303; 0.177); 0.608
DBCA	Cont.	−0.022 (−0.082; 0.037); 0.737	−0.022 (−0.075; 0.03); 0.251	−0.04 (−0.134; 0.055); 0.409	−0.012 (−0.053; 0.03); 0.577
	>LOD	−0.039 (−0.151; 0.074); 0.501	−0.032 (−0.244; 0.18); 0.766	0.051 (−0.085; 0.188); 0.454	−0.03 (−0.262; 0.202); 0.802
3-PBA	Cont.	−0.062 (−0.11; −0.013); 0.012	−0.012 (−0.041; 0.017); 0.494	−0.061 (−0.148; 0.026); 0.168	−0.041 (−0.114; 0.033); 0.277
	Q2	−0.088 (−0.301; 0.125); 0.409	−0.012 (−0.053; 0.03) 0.573	−0.038 (−0.179; 0.105); 0.607	−0.044 (−0.111; 0.024); 0.204
	Q3	−0.106 (−0.324; 0.112); 0.335	−0.015 (−0.048; 0.018); 0.368	−0.053 (−0.175; 0.071); 0.405	−0.043 (−0.192; 0.106); 0.572
	Q4	−0.083 (−0.142; −0.023); 0.007	−0.018 (−0.108; 0.072); 0.696	−0.045 (−0.259; 0.17); 0.684	−0.048 (−0.197; 0.102); 0.533

3PBA—3-phenoxybenzoic acid; CDCCA, TDCCA—cis- and trans-3-(2,2-Dichlorovinyl)-1-methylcyclopropane-1,2-dicarboxylic acid; DBCA—cis-2,2-dibromovinyl-2,2-dimethylcyclopropane-carboxylic acid. Q2—(25–50th) percentile; Q3—(50–75th) percentile; Q4—>75 percentile; reference category to Q2, Q3, Q4 is Q1; Q1—≤25th percentile; model adjusted for: SG, BMI, age, smoking, infertility diagnosis, AFC, and protocol type.

## Data Availability

Data available on request due to restrictions.

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
