# Peer review of "Synthetic Pyrethroids Exposure and Embryological Outcomes: A Cohort Study in Women from Fertility Clinic"

_ijerph, 2022, doi:10.3390/ijerph19095117_

Round 1

Reviewer 1 Report

The authors find with 450 women and 674 ART cycles a correlation between a synthetic pyrethroid, an insecticide, suspected as an endocrine-disrupting chemical, found in the urine, and the amount of mature oocytes.

Form

The article is adequate in length in all parts including tables and references. 27 % of the references are from the last five years.

Language is not bad but needs a professional correction.

The authors’ professional background should be given; is any of them a gynecologist subspecialized in reproductive medicine?

Content

For the clinician as a reader the positive findings should be given in absolute numbers: how many mature oocytes less if 3-PBA is how much too high? This should also be given in the abstract. (Statistical significance and clinical significance might be not the same…).

The average level of AMH in table 1 is not normal. Please comment on this and also please refer in this regard to your paper about the ovarian reserve in 2020.

After corrections the article should be published.

Author Response

Reviewer I.

The authors find with 450 women and 674 ART cycles a correlation between a synthetic pyrethroid, an insecticide, suspected as an endocrine-disrupting chemical, found in the urine, and the amount of mature oocytes.

Form

The article is adequate in length in all parts including tables and references. 27 % of the references are from the last five years.

Answer: Thank you.

Language is not bad but needs a professional correction.

Answer: The English grammar and language has been corrected by the native speaker but also by professional service (http://www.englishprep.pl/korekta.html) which correct the text based on the standards of The Society for Editors and Proofreaders (http://www.sfep.org.uk/).

The authors’ professional background should be given; is any of them a gynecologist subspecialized in reproductive medicine?

Answer: PaweÅ‚ Radwan, MichaÅ‚ Radwan  Renata Banaszczyk are the gynecologist working in Gameta fertility clinic. RafaÅ‚ KrasiÅ„ski – embryologist; Bartosz Wielgomas- responsible for exposure assessment, Anna Kilanowicz-Sapota-  involved data interpretation; Joanna Jurewicz - was involved in study concept, design and data interpretation.

Content

For the clinician as a reader the positive findings should be given in absolute numbers: how many mature oocytes less if 3-PBA is how much too high? This should also be given in the abstract. (Statistical significance and clinical significance might be not the same…).

Answer: As there is no reference value for the general population in the level of exposure to synthetic pyrethroids it is difficult to answer “how much is too high”. This is the first study on the effect of exposure to synthetic pyrethroids and embryological outcome. Little information is also about exposure to synthetic pyrethroids and fertility. In our study we have divided the exposure level of each pesticide metabolites as follow: 3-PBA urinary concentrations were categorized into quartiles (>25th to 50th , >50th to 75th , >75th), with the lowest quartiles considered as the reference group (<LOD to 25th ). The concentrations of CDCCA, TDCCA, DBCA were treated as < LOD (reference group) and > LOD due to low detection frequency 34.3%, 45% and 22.1% respectively. So it difficult to say that this amount is too high. Our level of exposure are similar with previously published paper regarding the level of exposure to synthetic pyrethroids.

We calculate the p-value to assess the statistical significance (p<0.007). There is less than a 0.7% probability the null is correct (no association). Therefore, we reject the null hypothesis, and accept the alternative hypothesis (3-PBA concentrations decrease MII oocyte count). In other words, there is sufficient evidence to conclude that there is an effect at the population level. The regression coefficients describe the mathematical relationship between each independent variable and the dependent variable (negative or positive).

The average level of AMH in table 1 is not normal. Please comment on this and also please refer in this regard to your paper about the ovarian reserve in 2020.

Answer: This was mistake the value was 11.2 ng/ml. 

After corrections the article should be published.

Answer: Thank you.

Reviewer 2 Report

The manuscript is original, well written but the conclusions are only partially supported. Therefore, I have the following minor comments.

 It is necessary to mention that the process may be affected by other substances.  In the environment are many EDs with similar effects and their concentration was also found in urine. So, pyrethroids are not the only suspected source of endocrine disruption. Therefore discussion lacks a potent argument for the choice of pyrethroids for monitoring. Please add.

The effect of pyrethroids on IVM in an animal pig model has been described. I miss the mention in the introduction and discussion.  Please add.

Author Response

Reviewer II.

The manuscript is original, well written but the conclusions are only partially supported. Therefore, I have the following minor comments.

Answer: Thank you.

 It is necessary to mention that the process may be affected by other substances.  In the environment are many EDs with similar effects and their concentration was also found in urine. So, pyrethroids are not the only suspected source of endocrine disruption. Therefore discussion lacks a potent argument for the choice of pyrethroids for monitoring. Please add.

Answer: The choice for monitoring pyrethroids was associated with the fact that there are little information about the level of exposure to synthetic pyrethroids in population. Because of the numerous applications and increasing worldwide use of pyrethroids more and more attention is being paid to the risk of exposure to humans. Pyrethroids have been linked to disruption of the endocrine system which can adversely affect reproduction, sexual development and interfere with the immune system in aminals. Compering to animal studies there is little information about the exposure to synthetic pyrethroids and health effects, especially reproductive effects.

According to our best knowledge this is the first study to assess environmental exposure to synthetic pyrethroids and embryological outcome among women undergoing IVF treatment. That is why the exposure to synthetic pyrethroids was analyzed. This information has been added to the Introduction Section.

The effect of pyrethroids on IVM in an animal pig model has been described. I miss the mention in the introduction and discussion.  Please add.

Answer: The reference has been added and the study has been described in Introduction and Discussion Section.

Reviewer 3 Report

Dear Authors,

This is an interesting study as synthetic pyrethoroids are popular and commonly used in agricultural, residential, public and veterinary areas for insect control. There are a few only epidemiological reports in the field, focused on associations between use of synthetic pyrethoroids and fertility. However, the manuscript needs a major revision before potential suitability for publication could be considered. Please find below my specific comments regarding the sections of the paper.

Tittle

  1. You might consider rephrasing the title of the manuscript (up to the authors)

Synthetic Pyrethroids Exposure Among Women from Fertility Clinic and Embryological Outcomes - is There an Association

change to:

Synthetic Pyrethroids Exposure and Embryological Outcomes: An Association Study In Women from Fertility Clinic

Abstract

  1. The aim of the study should match that formulated at the end of the Introduction section. Further, the objective should also contain information on study population, i.e. women undergoing fertility treatments.

Introduction

3. An explanation of the following term would be useful: early embryological outcomes. Introduction should be reworded and edited so that the reader are clearly informed about the rationale of the study: what was the background and motivation to conduct the study. The aim ought to be better grounded.

Materials and Methods

  1. Please order, revise and re-edit information in the following para: study desing, settings, participants.

More data would be expected about the department(s) / clinic(s), in which the data were collected, and by analogy, where these facilities were located. There is also a need of a reliable estimation what proportion of the women’s population nationwide is using services of these department(s).

  1. Regarding the following variables: demographics, medical and reproductive history, and lifestyle characteristics and occupational factors: a literature-based explanation would be necessary to explain such a selection, please cite appropriate literature data.

Results

  1. Table 3 needs a more accurate and informative comment.
  2. The title of the table 4 should be revised with an explicit information indicating „associations between ….”, and abbreviations should be explained below the table 4.

Discussion

  1. In the statements (lines 233-234), an appropriate citation and sources are needed.
  2. The phrases in lines 260-265 will require additional comments..
  3. The interpretation of the results should be extended.
  4. Strength and limitations need to be rethought and corrected.

General comments:

  1. Please explain abbreviations and meaning of the acronyms while first used in the text.
  2. There are several repetitions in the text that should be deleted, e.g.:
  • diagonal in line 205,
  • The statement in the lines 70-72 is duplicating another one in lines 230-232, and similarly the statement in lines 79-80 is repeated in the lines 245-246,
  • The expression „first study” several times (line 40, 86, 226, 266), whereas it could be sufficient to mention jus tonce in the strengths of the study (line 266),
  • Information about Bioethical Committee in lines 101-103 should be placed only in lines 296-298.

  1. The text requires typos correction (single spaces, foint size etc, e.g. line 205 and so on.
  2. The paper would be benefit from a English language editor`s input to help the readability of the work.
  3. Please put the changes with different color in the revised manuscript.
  4. Important: Please note that current version of the manuscript is too short regarding Journal’s style and requirement, which is preordained minimum as 3000 words. Thus, a modification and a careful revision of the text length is necessary after all the above points have been addressed.

Author Response

Reviewer III.

Dear Authors,

This is an interesting study as synthetic pyrethoroids are popular and commonly used in agricultural, residential, public and veterinary areas for insect control. There are a few only epidemiological reports in the field, focused on associations between use of synthetic pyrethoroids and fertility. However, the manuscript needs a major revision before potential suitability for publication could be considered. Please find below my specific comments regarding the sections of the paper.

 Tittle

  1. You might consider rephrasing the title of the manuscript (up to the authors)

Synthetic Pyrethroids Exposure Among Women from Fertility Clinic and Embryological Outcomes - is There an Association

 change to:

Synthetic Pyrethroids Exposure and Embryological Outcomes: An Association Study In Women from Fertility Clinic

 Answer: I have change the title according to the Reviewer suggestion to: Synthetic Pyrethroids Exposure and Embryological Outcomes: A Cohort Study In Women from Fertility Clinic”. I have delate “association study” this is in my opinion to use when epidemiological type of study is performed. In epidemiology we have cohort (prospective, retrospective), cross-sectional, case-control study. So it will not be proper to use “association study”. Association is the outcome not the type of study.

Abstract

  1. The aim of the study should match that formulated at the end of the Introduction section. Further, the objective should also contain information on study population, i.e. women undergoing fertility treatments.

Answer:  The aim of the study at the end of Abstract has been changed and now match that formulated at the end of the Introduction section.

  1. An explanation of the following term would be useful: early embryological outcomes. Introduction should be reworded and edited so that the reader are clearly informed about the rationale of the study: what was the background and motivation to conduct the study. The aim ought to be better grounded.

Answer: The rationale of the study: what was the background and motivation to conduct the study has been added according to the Reviewer suggestions and the Introduction has been re-written.

Materials and Methods

  1. Please order, revise and re-edit information in the following para: study desing, settings, participants.

More data would be expected about the department(s) / clinic(s), in which the data were collected, and by analogy, where these facilities were located. There is also a need of a reliable estimation what proportion of the women’s population nationwide is using services of these department(s).

Answer: All the information about the study population were described in details in our previously published study: Triclosan exposure and in vitro fertilization treatment outcomes in women undergoing in vitro fertilization. Environ Sc Poll Research 2021, 28,12993–12999. So that is why the population is described not in full details.

Study population: A total of 450 women between 25 and 45 years of age seeking infertility treatment at Gameta Hospital reproductive centre certified by the European Society for Human Reproduction and Embryology (ESHRE ART Center Certification for good clinical practice, 2019, C-0001) and that underwent at least one fresh in vitro fertilization cycle (n = 674 IVF-ICSI cycles) were recruited. The couples’ exclusion criteria were as follows: fertilization failure during the previous IVF-ICSI attempt, sperm concentration < 1 million per mL, azoopsermia, ovarian hyperstimulation syndrome, abnormal pelvic ultrasound examination (abnormal uterine cavity, hydrosalpinx, ovarian cysts), endocrinologic disorders (POCS (polycystic ovary syndrome)), menstrual disorders, chlamydia infection, thyroid disfunction (TSH > 2.5 μU/mL, BMI > 40 kg/m2 ). The Bioethical Committee in Lodz, Poland, approved the study (resolution no 23/2014). At the time of recruitment, study subjects received written informed consents before their participation and completed a questionnaire about sociodemographic characteristics, medical, especially gynecological history, chronic diseases, lifestyle factors, and occupational factors. The participant’s date of birth was collected at entry, and weight and height were measured by trained study staff. Body mass index (BMI) was calculated as weight (in kilograms) divided by height (in meters) squared. Clinical data assessment Participants’ clinical data were received from the medical electronic charts record. Concentration of hormones: folliclestimulating hormone (FSH), estradiol (E2), luteinizing hormone (LH), and progesterone, were assessed in serum using chemiluminescence immunoassay between second and third day of menstrual cycle. Serum was analyzed for antimüllerian hormone (AMH) with an enzyme-linked immunoabsorbent method utilizing commercially available Gen-II ELISA kits according to manufacturer instruction (Beckman Coulter, Inc., USA). The highest level of oestradiol prior to oocyte retrieval was treated as the peak oestradiol level.

  1. Regarding the following variables: demographics, medical and reproductive history, and lifestyle characteristics and occupational factors: a literature-based explanation would be necessary to explain such a selection, please cite appropriate literature data.

 Answer: These factors are well-known factors that can affect the results. These are the confounding factors that can impact on the association between exposure to pyrethroids and embryological outcomes. The selection of the factors was based on the literature that may affect the results. But it is a common procedure to collect the demographic characteristics and it is a part of each scientific study so in my opinion there is no need to find the support for such choice.

Results

  1. Table 3 needs a more accurate and informative comment.

Answer: The added not under the Table describe the Table in more details. Note: above diagonal diagonal - p values, Below– correlations; 3PBA- 3-phenoxybenzoic acid; CDCCA, TDCCA- cis- and trans- 3-(2,2-Dichlorovinyl)-1-methylcyclopropane-1,2-dicarboxylic acid; DBCA- cis-2,2-dibromovinyl-2,2-dimethylcyclopropane-carboxylic acid 

  1. The title of the table 4 should be revised with an explicit information indicating „associations between ….”, and abbreviations should be explained below the table 4.

 Answer: The title of the Table 4 has been changed. Abbreviations have been added under Table 4.

Discussion

  1. In the statements (lines 233-234), an appropriate citation and sources are needed.

Answer: This has been changed.

  1. The phrases in lines 260-265 will require additional comments..

Answer: The comments have been added.

  1. The interpretation of the results should be extended.

Answer: The interpretation of the study has been extended in case of animal studies. There is no human epidemiological studies on exposure to pyrethroids and female fertility which were not taken into account.

  1. Strength and limitations need to be rethought and corrected.

 Answer: The Strengths and limitations have been rewritten and corrected.

General comments:

  1. Please explain abbreviations and meaning of the acronyms while first used in the text.

Answer: All the abbreviations and meaning of the acronyms first used in the text have been explained.

  1. There are several repetitions in the text that should be deleted, e.g.:
  • diagonal in line 205,

Answer: This has been changed.

  • The statement in the lines 70-72 is duplicating another one in lines 230-232, and similarly the statement in lines 79-80 is repeated in the lines 245-246,

Answer: This has been changed.

  • The expression „first study” several times (line 40, 86, 226, 266), whereas it could be sufficient to mention jus tonce in the strengths of the study (line 266),

Answer: This has been changed.

  • Information about Bioethical Committee in lines 101-103 should be placed only in lines 296-298.

Answer: In my opinion this is a different context to use this information.

  1. The text requires typos correction (single spaces, foint size etc, e.g. line 205 and so on.

Answer: The text was checked according to Reviewer suggestions.

  1. The paper would be benefit from a English language editor`s input to help the readability of the work.

Answer: The English grammar and language has been corrected by the native speaker but also by professional service (http://www.englishprep.pl/korekta.html) which correct the text based on the standards of The Society for Editors and Proofreaders (http://www.sfep.org.uk/).

  1. Please put the changes with different color in the revised manuscript.

Answer: The revised version has been prepared using track changes.

  1. Important: Please note that current version of the manuscript is too short regarding Journal’s style and requirement, which is preordained minimum as 3000 words. Thus, a modification and a careful revision of the text length is necessary after all the above points have been addressed.

Answer: The article is now 3363 words.

Round 2

Reviewer 3 Report

To the Authors,

After checking the manuscript, I can see that, unfortunately, some of my comments have not been addressed. I would request to respond to the following points from Review 1:

  1. There is also a need of a reliable estimation what proportion of the women’s population nationwide is using services of Fertility Clinic.

  1. The sentences in lines 260-265 will require additional comments.

  1. There are several repetitions in the text that should be deleted, e.g.:
  • The statement in the lines 70-72 is duplicating another one in lines 230-232, and similarly the statement in lines 79-80 is repeated in the lines 245-246,
  • The expression „first study” several times (line 40, 86, 226, 266), whereas it could be sufficient to mention just once in the strengths of the study (line 266)

Author Response

After checking the manuscript, I can see that, unfortunately, some of my comments have not been addressed. I would request to respond to the following points from Review 1:

  1. There is also a need of a reliable estimation what proportion of the women’s population nationwide is using services of Fertility Clinic.

 Answer: The patients enrolled  for the study  were form the area of Lodz agglomeration. The population of Lodz agglomeration  is about 1 million inhabitants. According to European IVF Monitoring Consortium.  The demand for IVF in Poland is  about 600- 700 cycles for one million people. The size of  infertile population (age 20-40) in Lodz area is estimated to 10.000 couples.  Seven hundred  couples  needs IVF treatment per year.  Our group corresponds to about 64 % of this population

Taking to consideration all country  according to European IVF Monitoring  1560 IVF cycles are made in Poland.  The enrolled groups corresponds  to   43 %

References: European IVF-Monitoring Consortium (EIM) for the European Society of Human Reproduction and Embryology (ESHRE) , Wyns C, De Geyter C, et al. ART in Europe, 2017: results generated from European registries by ESHRE. Hum Reprod Open. 2021;2021(3):hoab026. Published 2021 Aug 5. doi:10.1093/hropen/hoab026

  1. The sentences in lines 260-265 will require additional comments.

 Answer: In the sentence the additional comments have been added.

  1. There are several repetitions in the text that should be deleted, e.g.:
  • The statement in the lines 70-72 is duplicating another one in lines 230-232, and similarly the statement in lines 79-80 is repeated in the lines 245-246,

Answer: The duplicate sentences have been delated.

  • The expression „first study” several times (line 40, 86, 226, 266), whereas it could be sufficient to mention just once in the strengths of the study (line 266)

Answer: This has been changed.